# Perturbation Deterioration: The Other Side of Catastrophic Overfitting

## Abstract

Our goal is to understand *why the robustness accuracy would abruptly drop to zero, after conducting FGSM-style adversarial training for too long*. While this phenomenon is commonly explained as overfitting, we observe that it is a twin process: not only does the model catastrophic overfits to one type of perturbation, but also the perturbation deteriorates into random noise. For example, at the same epoch when the FGSM-trained model catastrophically overfits, its generated perturbations deteriorate into random noise. Intuitively, once the generated perturbations become weak and inadequate, models would be misguided to overfit those weak attacks and fail to defend strong ones. In the light of our analyses, we propose APART, an adaptive adversarial training method, which parameterizes perturbation generation and progressively strengthens them. In our experiments, APART successfully prevents perturbation deterioration and catastrophic overfitting. Also, APART significantly improves the model robustness while maintaining the same efficiency as FGSM-style methods, e.g., on the CIFAR-10 dataset, APART achieves 53.89% accuracy under the PGD-20 attack and 49.05% accuracy under the AutoAttack[1].

## 1 Introduction

While neural networks keep advancing the state of the arts, their vulnerability to adversarial attacks (Szegedy et al., 2013) casts a shadow over their applications—subtle, human-imperceptible input shifts can fool these models and alter their predictions. Adversarial training adds perturbations to model inputs during training, and is one of the most successful approaches to establish model robustness (Goodfellow et al., 2014; Madry et al., 2017; Kurakin et al., 2017; Tramèr et al., 2017; Zhang et al., 2019b; Liu et al., 2018).

One crucial and common challenge in adversarial training is the significant computation overhead, e.g., it may take 3-30 times longer to conduct adversarial training than the vanilla training. In response to this challenge, there has been a recent surge in work aiming to reduce the computation overhead (Goodfellow et al., 2014; Shafahi et al., 2019a; Zhang et al., 2019a; Wong et al., 2020). Although these methods successfully accelerates adversarial training, they lead to an unexpected phenomenon—when conducting FGSM-style adversarial training, the model robustness would abruptly drop to zero (Rice et al., 2020) at certain epoch. This phenomenon is referred as *catastrophic overfitting*, and the robustness drop is usually viewed as model overfitting—the model overfits to one specific type of perturbation (Rice et al., 2020; Wong et al., 2020; Kim et al., 2021).

Here, we show that the robustness drop is a twin process. Besides the model overfitting to one type of perturbation, we observe perturbations becoming too weak to establish model robustness. For example, as visualized in Figure 1, both the model and the perturbation change dramatically between Epoch 15 and Epoch 16—the robust accuracy of the model drops from 45 to almost zero, the perturbation strength deteriorates dramatically. As implied by this phenomenon, we suggest that there exists strong correlations between the catastrophic overfitting and perturbation deterioration. Intuitively, without perturbation deterioration, even if the model overfits to one type of perturbation with a reasonable strength, it would lead to a sub-optimal robustness instead of entirely diminished robust accuracy. Meanwhile, once the perturbation deteriorates into random noise, overfitting to that random noise-like perturbation could cause the model robustness drop to zero.

---

[1]Code will be released under the Apache-2.0 license for future studies.

Table 1: Notation Table (Elaborated in Section 2)

| **x** is input | $y$ is label | $\alpha$ is step size | $\mathcal{L}$ is objective | $\Delta \mathbf{x} = \partial \mathcal{L} / \partial \mathbf{x}$ | $\Delta \theta = \partial \mathcal{L} / \partial \theta$ |
|---|---|---|---|---|---|
| | $\theta_{\mathcal{A}}^{(i)}$ | model parameter $\theta$ trained for $i$ epochs by method $\mathcal{A}$ | | | |
| | $f_{\mathcal{A}}(\theta, \mathbf{x}, y)$ | perturbation generated by method $\mathcal{A}$ to attack model $\theta$ on $\mathbf{x}$, $y$ | | | |
| | $\omega_{\mathbf{x}}$ | perturbation initialization as parameterized by FGSM+ and APART | | | |
| | $\mathcal{G}_{\mathcal{B},i}(\mathcal{A})$ | perturbation strength of $f_{\mathcal{A}}(\theta_{\mathcal{A}}^{(i)})$, calculated as its gap to $f_{\mathcal{B}}(\theta_{\mathcal{A}}^{(i)})$ on model $\theta_{\mathcal{A}}^{(i)}$ | | | |
| $Acc(\theta_{\mathcal{A}}^{(i1)}, f_{\mathcal{B}}(\theta_{\mathcal{C}}^{(i2)}, \cdot))$ | | accuracy of $\theta_{\mathcal{A}}^{(i1)}$ when attacked by perturbations generated by $f_{\mathcal{B}}(\cdot)$ for $\theta_{\mathcal{C}}^{(i2)}$ | | | |

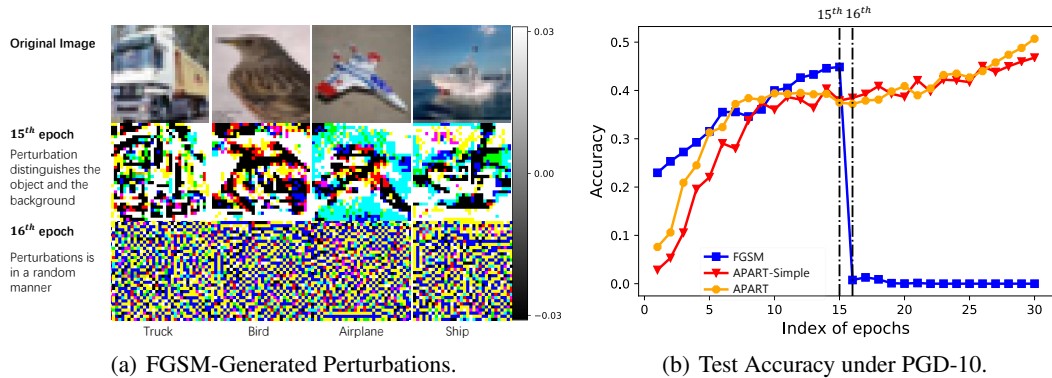

(a) FGSM-Generated Perturbations.  (b) Test Accuracy under PGD-10.

Figure 1: Analyses of FGSM-generated perturbations (for Pre-ResNet18 on the CIFAR-10 dataset).

From this perspective, the key to prevent catastrophic overfitting falls upon shielding models from perturbation deterioration. Correspondingly, we design an adaptive adversarial training method, APART, which parameterizes perturbation generation and updates its parameters to progressively strengthen the perturbation generator with gradient ascent. Specifically, we first treat perturbation initialization as parameters—instead of starting from scratch every time, APART improve the initialization with gradient ascent. Then, APART factorize the input perturbation as a series of perturbations, which are integrated with learnable step sizes and can self-adapt to different scenarios. In our experiments, APART leads to consistent performance improvements over FGSM-style algorithms while maintaining roughly the same efficiency.

## 2 PRELIMINARIES AND NOTATIONS

Given a neural network with $n$ convolution blocks, we denote the input of the $i$-th block as $\mathbf{x}_i$, the input and output of the entire network as $\mathbf{x}$ and $y$. Note that, $\mathbf{x}$ and $\mathbf{x}_1$ are the same in conventional residual networks, such as ResNet (He et al., 2016a), Wide ResNet (Zagoruyko & Komodakis, 2016), and Pre-Act ResNet (He et al., 2016b). Adversarial training aims to establish the model robustness by solving the optimization problem as below ($\theta$ is the network parameter, $\delta$ is the perturbation, $(\mathbf{x}, y)$ is a data-label pair, and $f(\cdot)$ is the perturbation generation function).

$$\min_{\theta} \mathcal{L}(\theta, \mathbf{x} + \delta, y) \ s.t. \ \delta = f(\theta, \mathbf{x}, y). \tag{1}$$

Different adversarial training methods generate perturbations differently. Ideally, adversarial training should use the most effective (i.e., strongest) perturbation within the same norm constraint, i.e., $f^*(\cdot) = \text{argmax}_{||\delta|| \le \epsilon} \mathcal{L}(\theta, \mathbf{x} + \delta, y)$. In practice, as an approximation, $f(\cdot)$ is typically implemented as gradient ascent with fixed iteration and step size. For example, the FGSM algorithm directly calculates perturbations as $f_{\text{FGSM}}(\theta_{\text{FGSM}}, \mathbf{x}, y) = \epsilon \cdot \text{sign}(\Delta \mathbf{x})$. Also, we use $\theta_{\text{method}}^{(i)}$ to refer to model parameters that are trained by a specific adversarial training algorithm for $i$ epochs. We use $Acc(\theta_{\mathcal{A}}^{(i1)}, f_{\mathcal{B}}(\theta_{\mathcal{C}}^{(i2)}, \cdot))$ to indicate the performance of model $\theta_{\mathcal{A}}^{(i1)}$, under $f_{\mathcal{B}}(\theta_{\mathcal{C}}^{(i2)}, \cdot)$, i.e., the perturbation generated by method $f_{\mathcal{B}}(\cdot)$ to attack $\theta_{\mathcal{C}}^{(i2)}$. These notations are summarized in Table 1.

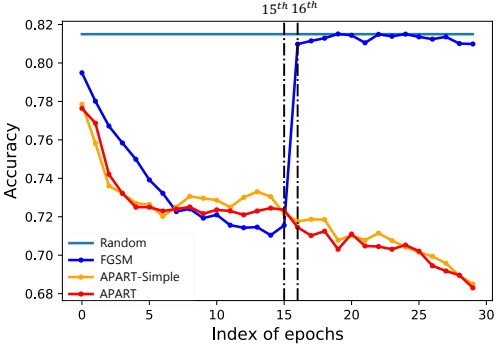 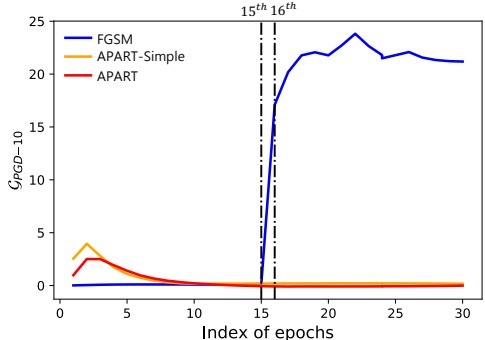

(a) Accuracy of $\theta^{(30)}_{\text{PGD-10}}$ on transferred attacks (smaller value indicates stronger attacks). Line named as method refers to $Acc(\theta^{(30)}_{\text{PGD-10}}, f_{\text{FGSM}}(\theta^i_{\text{method}}, \cdot))$, while Random refers to $Acc(\theta^{(30)}_{\text{PGD-10}}, f_{\text{random}}(\cdot))$.

(b) Strength gap between method and PGD-10, i.e., $\mathcal{L}(\theta^{(i)}_{\text{method}}; \mathbf{x} + f_{\text{method}}(\theta^{(i)}_{\text{method}}, \mathbf{x}, y), y) - \mathcal{L}(\theta^{(i)}_{\text{method}}; \mathbf{x} + f_{\text{PGD-10}}(\theta^{(i)}_{\text{method}}, \mathbf{x}, y), y)$. Smaller value indicates stronger attacks.

Figure 2: Perturbation strength in different epochs. In part (a), perturbation strength is estimated by transfer the adversary image to attack a model trained by PGD-10 separately for 30 epochs. In part (b), perturbation strength is estimated as the gap to a stronger attack (PGD-10 here)

## 3 PERTURBATION DETERIORATION AND CATASTROPHIC OVERFITTING

Typically, the robustness drop is viewed as model overfitting (Rice et al., 2020; Wong et al., 2020; Kim et al., 2021) and this phenomenon is referred as catastrophic overfitting. Meanwhile, it has been observed that catastrophic overfitting only happens to simple methods like FGSM (Goodfellow et al., 2014) [2], indicating this phenomenon is not only about model overfitting, but also the strength of perturbation. Inspired by this observation, we aim to explore the other side of catastophic overfitting, i.e., *perturbation deterioration*.

### 3.1 PERTURBATION STRENGTH

To verify our intuition, we try to empirically estimate the perturbation strength and analyze its dynamics during the adversarial training.

First, we try to estimate the perturbation strength of $f_{\mathcal{A}}$ as $Acc(\theta^{30}_{\text{PGD-10}}, f_{\mathcal{A}}(\theta^i_{\mathcal{A}}, \cdot))$, i.e., using $f_{\mathcal{A}}(\theta^i_{\mathcal{A}})$ to attack a model trained separately with PGD-10 for 30 epochs. For comparisons, we also list the performance of $\theta^{30}_{\text{PGD-10}}$ under random noise. As visualized in Figure 2(a), in the first 15 epochs, the perturbation strength of FGSM keeps increasing, while at the 16 epoch, its strength dramatically drops to the random noise level. This shows that, after the perturbation deterioration, the perturbations not only look like random noise, but also behaves like random noise.

Alternatively, we estimate the perturbation strength of $f_{\mathcal{A}}$ as its gap to a more powerful method $f_{\mathcal{B}}$. Specifically, we calculate the strength for $f_{\mathcal{A}}(\theta^{(i)}_{\mathcal{A}}, \cdot)$ as $\mathcal{G}_{\mathcal{B},i}(\mathcal{A})$ in Equation 2. Intuitively, the weaker the perturbation $f_{\mathcal{A}}$ is, the larger $\mathcal{G}_{\mathcal{B},i}(\mathcal{A})$ would be.

$$\mathcal{G}_{\mathcal{B},i}(\mathcal{A}) = \mathcal{L}(\delta_{\mathcal{A}}) - \mathcal{L}(\delta_{\mathcal{B}}) \text{ where } \mathcal{L}(\delta_{\text{method}}) = \mathcal{L}(\theta^{(i)}_{\mathcal{A}}; \mathbf{x} + f_{\text{method}}(\theta^{(i)}_{\mathcal{A}}, \mathbf{x}, y), y). \quad (2)$$

We conduct experiments with Pre-Act ResNet18 on the CIFAR-10 dataset and visualize $\mathcal{G}_{\text{PGD-10},i}(\text{FGSM})$ in Figure 2(b). It shows that the strength gap between PGD-10 and FGSM is small in the early stage, dramatically explodes at the 16th epoch, and keeps a large value since then (i.e., the perturbation strength mostly vanishes at the 16th epoch).

**Perturbation Deterioration.** In both cases, the timing of perturbation strength deterioration coincides with the timing of the robustness drop, thus supporting our intuition that the other side of

---

[2]For a FGSM trained Pre-ResNet18 (on CIFAR-10), its accuracy under PGD-20 attacks drops from 45% (epoch 15) to 0% (epoch 16); For a PGD-10 trained model, it drops from 49% (epoch 30) to 38% (epoch 200).

---

**Algorithm 1:** APART (the first and the second round propagations are marked with red and blue; $\epsilon$ is the perturbation bound; $\mu_\theta$, $\mu_\omega$, and $\mu_\alpha$ are learning rates for $\theta$, $\omega$, and $\alpha$; Table 1 summarizes others notations).

---

1 **while** not converged **do**
2      **for** $\mathbf{x}, y$ in the training set **do**
3          $\delta_1 \leftarrow \alpha_\omega \cdot \omega_\mathbf{x}$ //initialize the perturbation for the model input.
4          $\delta_1 \leftarrow \max(\min(\delta_1 + \alpha_1 \cdot \mathrm{sign}(\frac{\partial\mathcal{L}(\theta;\mathbf{x}+\delta_1,y)}{\partial\mathbf{x}_1}),-\epsilon),+\epsilon)$ //calculate $\delta_1$.
5          **for** each residual block with index $i > 1$ **do**
6              $\delta_i \leftarrow \alpha_i \cdot \mathrm{sign}(\frac{\partial\mathcal{L}(\theta;\mathbf{x}+\delta_1,y)}{\partial\mathbf{x}_i})$ //calculate perturbations for block i.
7          $\theta = \theta - \mu_\theta \cdot \frac{\partial\mathcal{L}(\theta;\{\mathbf{x}_i\}_{i=1}^n+\{\delta_i\}_{i=1}^n,y)}{\partial\theta}$ //update parameters.
8          $\omega_\mathbf{x} = \max(\min(\omega_\mathbf{x} + \mu_\omega \cdot \mathrm{sign}(\frac{\partial\mathcal{L}(\theta;\{\mathbf{x}_i\}_{i=1}^n+\{\delta_i\}_{i=1}^n,y)}{\partial\omega_\mathbf{x}}),-1),1)$ //update $\omega_\mathbf{x}$.
9          $\alpha_i = \alpha_i + \mu_\alpha \cdot (\frac{\partial\mathcal{L}(\theta;\{\mathbf{x}_i\}_{i=1}^n+\{\delta_i\}_{i=1}^n,y)}{\partial\alpha_i} - \lambda \cdot \frac{\partial|\alpha_i|_2^2}{\partial\alpha_i})$ //update step sizes.
10 **return** $\theta$

---

catastrophic overfitting is perturbation deterioration. Intuitively, adversarial training cannot establish satisfying model robustness without strong enough perturbations, and strong perturbations require a small gap $\mathcal{G}(\cdot)$. However, one local optima of Equation 1 is the parameter $\theta$ that deteriorates $f(\theta, \cdot)$ into random noise, which advances the optimization of Equation 1 at the cost of deteriorated perturbations. Also, since all parameter updates are made to decrease $\mathcal{L}$, most existing methods have no regularization to keep the perturbation strength.

### 3.2 ADAPTIVITY HELPS PREVENT CATASTROPHIC OVERFITTING

Here, we further verify our intuition by showing that it prevents the catastrophic overfitting by alleviating the perturbation deterioration. Intuitively, one straightforward way to strengthen generators is to parameterize them and update them together with model parameters. Specifically, we treat the perturbation initialization for the input (denoted as $\omega_\mathbf{x}$) and the step size (referred as $\alpha$) as parameters of FGSM, and change the objective from Equation 1 to:

$$\max_{\alpha,\omega} \min_\theta \mathcal{L}(\theta; \mathbf{x} + f_{\mathrm{FGSM}}(\alpha, \omega_\mathbf{x}; \theta, \mathbf{x}, y), y). \tag{3}$$

During model training, we update $\theta$ with gradient descent and update $\alpha$ and $\omega_\mathbf{x}$ with gradient ascent. We refer this variant as **APART-Simple**. Note that its only difference to FGSM is that APART-Simple is parameterized and can be enhanced during training, thus suffering less from perturbation deterioration.

We conduct experiments with Pre-Act ResNet18 on the CIFAR-10 dataset, and visualize $Acc(\theta_{\mathrm{PGD\text{-}10}}^{(30)}, f_{\mathrm{FGSM}}(\theta_{\mathrm{APART\text{-}Simple}}^{(i)}, \cdot))$ in Figure 2(a) and $\mathcal{G}_{\mathrm{PGD\text{-}10},i}(\mathrm{APART\text{-}Simple})$ in Figure 2(b). It shows that APART-Simple does not suffer from the catastrophic overfitting. This not only supports our intuition that the perturbation deterioration is one cause of the catastrophic overfitting, but motivates us to add more adaptivity to the perturbation generator.

## 4 ADAPTIVE ADVERSARIAL TRAINING

Guided by our analyses, we propose to improve FGSM by further improving the perturbation generator during the training. Since the algorithm features the ability to adapt itself, we refer our method as adaptive adversarial training (APART). Specifically, it first factorizes the perturbation for the input image into a series of perturbations, one for each residual block. Moreover, it employ different step sizes for different perturbations, treat them as learnable parameters, update them to integrate perturbations adaptively, and strengthen the generator during the model training.

**Factorize the Input Perturbation.** For a multi-layer network, the perturbation generated at the input attacks not only the first layer, but also all following layers. Intuitively, existing methods like PGD-N implicitly blender these attacks with N additional forward- and backward-propagations on

Table 2: Model Performance of WideResNet34-10 on the CIFAR-10 dataset.

| Efficient | Methods | PGD-20 | AA | C&W | Gaussian | Clean Data | Time/Epoch |
|:---:|:---|:---|:---|:---|:---|:---|:---|
| ✗ | ATTA-10 | 54.33% | 49.10% | **59.11%** | 77.05% | 83.80% | 706 secs |
| ✗ | PGD-10 | **55.41%** | **52.08%** | 58.77% | **77.70%** | **86.43%** | **680 secs** |
| ✓ | Free-8 | 47.68% | 46.21% | 56.31% | 75.98% | **85.54%** | 252 secs |
| ✓ | S+FGSM | 36.71% | 33.15% | 44.50% | **81.25%** | 85.15% | 232 secs |
| ✓ | F+FGSM | 46.37% | 44.27% | 56.21% | 75.10% | 85.10% | **122 secs** |
| ✓ | APART | **53.89%** | **49.05%** | **58.50%** | 77.31% | 84.65% | 162 secs |

the input perturbation, which significantly inflates the computation cost. Here, we factorize the input perturbation as a series of perturbations and explicitly learn to combine them. Specifically, we refer to the input of $i$-th residual block as $\mathbf{x}_i$ and the output of $i$-th residual block as $\mathbf{x}_i +$ CNNs$(\mathbf{x}_i)$. Then, we add the perturbation $\Delta\mathbf{x}_i$ to the input of CNNs$(\cdot)$ to establish its robustness, *i.e.*, $\mathbf{x}_i + $CNNs$(\mathbf{x}_i + \delta_i)$ where $\delta_i = \alpha_i \Delta\mathbf{x}_i$. Similar to PGD-N, this approach also involves multiple perturbations; different from PGD-N, these perturbations can be calculated with the same forward- and backward-propagations.

**Initialization Parameterization.** Similar to APART-Simple, we also treat perturbation initializations as learnable parameters to better defense the deterioration. Since it consumes additional memory, we only parameterize the perturbation initialization at the input, and keep all other perturbations zero-initialized. In this way, the additional storage has roughly the same size with the dataset.

**APART Algorithm.** We summarize APART in Algorithm 1. Same with FGSM, it contains two rounds of forward- and backward-propagations. In the first round, it initializes the input perturbation and calculates gradients for both the input of the first layer and the input of all the following blocks. In the second round, it applies the generated perturbations to the input of the corresponding blocks, *i.e.*, change $\mathbf{x}_i + $CNNs$(\mathbf{x}_i)$ to $\mathbf{x}_i + $CNNs$(\mathbf{x}_i + \delta_i)$. Then, besides updating model parameters with gradient descent, we enhance the generator with gradient ascent (*i.e.*, updating step sizes $\alpha_i$ and the perturbation initialization $\omega_\mathbf{x}$). Note that, to control the magnitude of step sizes $\alpha_i$, we add a $L_2$ regularization to its updates and use $\lambda$ to control it (as line 9 in Algorithm 1).

Note that, calculating the exact gradients of $\omega_\mathbf{x}$ or $\alpha_\omega$ requires a second order derivation ($\Delta\omega_\mathbf{x}$ and $\Delta\alpha_\omega$ are based on $\delta_i$, and the calculation of $\delta_i$ includes some first order derivations involving $\omega_\mathbf{x}$ and $\alpha_\omega$). Due to the success of the First-order MAML (FOMAML) (Finn et al., 2017), we simplifies the calculation by omitting higher order derivations. Specifically, FOMAML demonstrates the effectiveness to ignore higher order derivations and approximate the exact gradient with only first order derivations. Here, we have a similar objective with FOMAML—FOMAML aims to find a good model initialization, and we try to find a good perturbation initialization. Thus, we also restrict gradient calculations to first-order derivations. In this way, APART has roughly the same computation complexity with FGSM and it is significantly faster than PGD-N.

## 5 EXPERIMENTS

As in Figures 2(a) and Figure 2(b), APART shields adversarial training from catastrophic overfitting and largely alleviates the robustness drop. Systematic evaluations are further conducted as below.

### 5.1 EXPERIMENTAL SETTINGS

**Datasets.** We conduct experiments on the CIFAR-10 and CIFAR-100 datasets (Krizhevsky, 2009) as well as the ImageNet dataset (Krizhevsky et al., 2012).

**Neural Architectures.** We conduct experiments with ResNet He et al. (2016a), Pre-ResNet (He et al., 2016b) and WideResNet (Zagoruyko & Komodakis, 2016). Specifically, we use Pre-ResNet18 and WideResNet34-10 on the CIFAR-10 dataset, Pre-ResNet18 on the CIFAR-100 dataset, and ResNet50 (He et al., 2016a) on the ImageNet dataset.

**Data Augmentation.** Following the previous work, we apply the standard data augmentation. For the CIFAR datasets, we apply random flipping as a data augmentation procedure and take a random

Table 3: Model Performance of Pre-ResNet18 on the CIFAR-10 dataset.

| Efficient | Methods | PGD-20 | AA | C&W | Gaussian | Clean Data | Time/Epoch |
|:---:|:---|:---:|:---:|:---:|:---:|:---:|:---:|
| $\times$ | ATTA-10 | 49.03% | 45.70% | 58.30% | **73.10%** | **82.10%** | 140 secs |
| $\times$ | PGD-10 | **52.16%** | **48.50%** | **58.97%** | 72.33% | 82.05% | **133 secs** |
| $\checkmark$ | Free-8 | 47.37% | 44.53% | 56.10% | 72.56% | 81.64% | 62 secs |
| $\checkmark$ | S+FGSM | 34.47% | 32.15% | 42.21% | 73.45% | **89.28%** | 61 secs |
| $\checkmark$ | F+FGSM | 46.06% | 42.37% | 55.34% | 72.25% | 83.81% | **20** secs |
| $\checkmark$ | APART | **51.30%** | **45.92%** | **58.73%** | **73.65%** | 82.80% | 29 secs |

crop with $32 \times 32$ from images padded by 4 pixels on each side (Lee et al., 2015). For the ImageNet dataset, we divide the training into three phases, where phases 1 and 2 use images resized to 160 and 352 pixels and the third phase uses the original images (Wong et al., 2020).

**Optimizer.** For all experiments, we use SGD with momentum as the optimizer. The momentum factor is set to 0.9 and the training is conducted for 60 epochs with cyclic learning rate (Smith, 2017), where the maximum learning rate is set to 0.2 and minimum learning rate is set to 0. For the ImageNet dataset, we adopt a setting similar to Wong et al. (2020) and train the model for 15 epochs; The maximum learning rate of cyclic learning rate schedule is set to 0.4 and the learning rate is set to 0.

**Other Hyper-parameters.** For all experiments, we apply the cyclic learning rate scheduler for $\mu_\alpha$. On the CIFAR datasets, the maximum learning rate and $\lambda$ are set as $5 \times 10^{-8}$ and 200, respectively; on the ImageNet dataset, they are set as $4 \times 10^{-9}$ and 5000, respectively. Due to the similarity between line 4 and line 7 in Algorithm 1, we set $\mu_\omega$ as $\frac{\alpha_1}{\alpha_\omega}$, which makes the update on $\omega$ has a similar impact with the update in line 4 of Algorithm 1.

**Robustness Evaluation.** We adopt PGD-20, AutoAttack (Croce & Hein, 2020), Gaussian random noise, and C&W (Carlini & Wagner, 2017) as the attack methods for evaluation. For both adversarial training and evaluation, we restrict perturbations to $|\delta|_\infty \leq 8/255$ on the CIFAR datasets, $|\delta|_\infty \leq 2/255$ on the ImageNet dataset.

**Infrastructure.** Our experiments are conducted with NVIDIA Quadro RTX 8000 GPUs; mixed-precision arithmetic (Micikevicius et al., 2018) is adopted to accelerate model training; the training speed of APART or baselines is evaluated on an idle GPU.

## 5.2 COMPARED METHODS

For comparison, we select three state-of-the-art adversarial training methods, which features efficient training. Also, we list two other adversarial training methods that are significant slower. On the CIFAR datasets, we report accuracy and training time based on our experiments. As to ImageNet, we directly refer to the number reported in the original papers.

- **PGD-N** (Madry et al., 2017) is a classical, sophisticated adversarial training method. PGD is an iterative version of FGSM with uniform random noise as initialization and N is the number of iterations.
- **ATTA-K** (Zheng et al., 2019) uses the adversarial examples from neighboring epochs. K is the number of iterations and denotes the strength of attack.
- **Free-m** (Shafahi et al., 2019a) uses the same backward propagation to update both the model and trains on the same minibatch $m$ times in a row. Here we set $m = 8$.
- **F+FGSM** (Wong et al., 2020) uses a large step size and random initialization to improve FGSM. It achieves comparable performance with PGD-10, but still suffers from the robustness drop.
- **S+FGSM** (Kim et al., 2021) uses several checkpoints to validate the inner interval of perturbation direction to determine the appropriate magnitude of the perturbation of each image.

## 5.3 PERFORMANCE COMPARISON

Generally, we observe that PGD and ATTA achieves better performance than other methods, at the cost of significant training overheads. Meanwhile, APART achieves consistent performance improvements over FGSM-style methods, while maintaining roughly the same training speed.

Table 4: Model Performance of Pre-ResNet18 on the CIFAR-100 dataset.

| Efficient | Methods | PGD-20 | AA | C&W | Gaussian | Clean Data | Time/Epoch |
|:---:|:---|:---:|:---:|:---:|:---:|:---:|:---:|
| × | ATTA-10 | 25.60% | 22.90% | 30.75% | 42.10% | 56.20% | 140 secs |
| × | PGD-10 | **28.10%** | **25.11%** | **33.35%** | **42.41%** | **57.23%** | **133 secs** |
| ✓ | Free-8 | 25.88% | 22.15% | 30.55% | 42.15% | 55.13% | 62 secs |
| ✓ | S+FGSM | 10.15% | 8.91% | 15.11% | **50.00%** | **71.10%** | 61 secs |
| ✓ | F+FGSM | 25.31% | 21.32% | 30.06% | 42.03% | 57.95% | **20 secs** |
| ✓ | APART | **27.56%** | **23.38%** | **32.36%** | 44.37% | 58.40% | 29 secs |

Table 5: Model Performance of ResNet50 on the ImageNet dataset. $^+$ indicates single precision training.

| Efficient | Methods | Clean Data | PGD-10 Attack | Time/Epoch |
|:---:|:---|:---:|:---:|:---:|
| × | ATTA-2 (Zheng et al., 2019) | 60.70% | 44.57% | $4.85^+$ hrs |
| ✓ | Free-4 (Shafahi et al., 2019a) | 64.44% | 43.52% | 3.46 hrs |
| ✓ | F+FGSM (Wong et al., 2020) | 60.90% | 43.46% | 0.8 hrs |
| ✓ | APART | 60.52% | 44.30% | 1 hrs |

Table 6: Ablation study of APART on the CIFAR-10 dataset with Pre-ResNet20.

| Training Methods | Clean Data | PGD-20 Attack | AA |
|:---|:---:|:---:|:---:|
| APART | 82.80% | 51.30% | 45.92% |
| APART (w/o layer-wise perturbation) | 83.64% | 47.28% | 43.10% |
| APART (w/o perturbation initialization) | 82.42% | 50.10% | 45.10% |

Specifically, we summarize results on the CIFAR-10 in Table 2 and Table 3, CIFAR-100 in Table 4, and ImageNet in Table 5. Comparing to Free-8, S+FGSM, and F+FGSM, APART achieves consistent performance improvements against PGD-20, AutoAttack, and C&W attack for both Pre-ResNet18 and WideResNet34-10. As to ATTA-10 and PGD-10, APART achieves slightly worse performance with 4+ times speedup. Among all methods, F+FGSM is the fastest method, and APART significantly improves the model robustness without significant computation overheads. For example, F+FGSM takes 122 secs/epoch for training WideResNet34-10 on the CIFAR-10 dataset, and APART takes 162 secs/epoch to achieve a 7.52 absolute accuracy improvement under the PGD-20 attack and a 4.78 absolute accuracy improvement under the AutoAttack.

It is worth mentioning that, on CIFAR-10 and CIFAR-100, S-FGSM has the best performance under Gaussian noise attack due to its high clean accuracy while the performances under PGD-20 and C&W attack are much poorer than other methods. This trade-off is further discussed in Section 5.4.

## 5.4 BALANCING CLEAN ACCURACY AND ROBUST ACCURACY

As shown in Tables 3, 2, and 4, the robustness improvement usually comes at the cost of accuracy on clean images. Meanwhile, the performances on corrupted images have consistent trends, *e.g.*, if method $\mathcal{A}$ outperforms $\mathcal{B}$ under PGD-20 attack, $\mathcal{A}$ likely also outperforms $\mathcal{B}$ under AutoAttack or C&W attack. To better understand the trade-off between the clean accuracy and the model robustness, we employ different $\epsilon$ values during training (i.e., $[\frac{2}{255}, \frac{3}{255}, \cdots, \frac{10}{255}]$), train multiple models for each method, and visualize their performance of Pre-ResNet20 on the CIFAR-10 dataset in Figure 4. Points of APART locate in the right top corner of the figure and significantly outperform other methods. This further verifies the effectiveness of APART.

## 5.5 ABLATION STUDIES

APART employs two techniques to parameterize the perturbation generator. The first is to learn an initialization for the perturbation, and the second is to factorize input perturbations into a series of

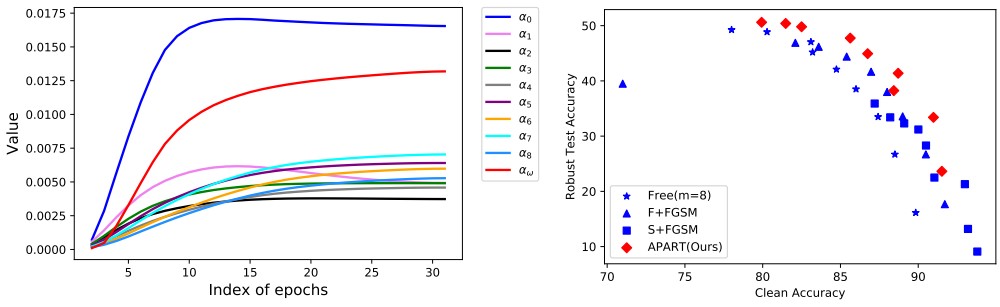

Figure 3: Step sizes in different epochs (with Pre-ResNet18 on CIFAR-10)

Figure 4: Pre-ResNet18 performance on CIFAR-10 ($\epsilon_{train} = [\frac{2}{255}, \cdots, \frac{10}{255}]$)

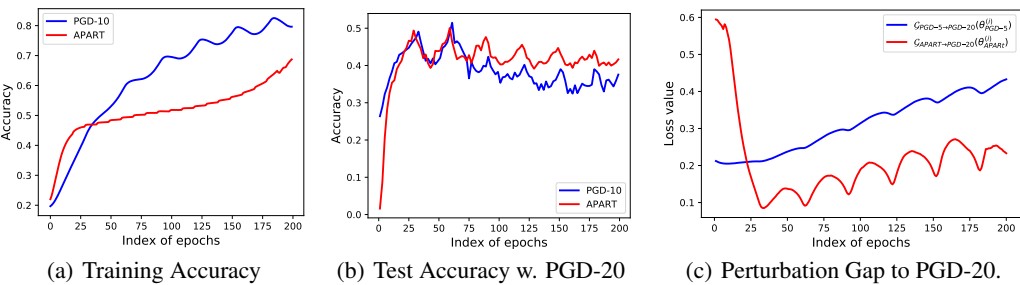

(a) Training Accuracy     (b) Test Accuracy w. PGD-20     (c) Perturbation Gap to PGD-20.

Figure 5: APART can alleviate the robust overfitting (note that APART is $\sim 4$x faster than PGD-10).

perturbations. To understand the effectiveness of them, we conduct an ablation study and summarized the results in Table 6. Removing layer-wise perturbations leads to a 4.02% drop and a 2.48% drop on accuracy under PGD-20 attack and AutoAttack respectively; Removing perturbation initialization leads to a 1.20% drop and a 0.82% drop on accuracy under PGD-20 attack and AutoAttack respectively. Therefore, both techniques are helpful and necessary to achieve a better model robustness.

## 5.6 Evolution of Step Sizes

After factorizing the input perturbation as a series of perturbations, we employ learnable step sizes to compose perturbations more effectively. To better understand these step sizes, we visualize their values during the training of Pre-ResNet20 on the CIFAR-10 dataset in Figure 3. It shows that the first-layer perturbation is more important than others. Also, the step sizes of perturbation at the first and the second layers decrease after 10 epochs, while other step sizes keep increasing across the training. This phenomenon verifies that APART is able to adapt the generator setting to different training stages.

## 5.7 Alleviation of Robust Overfitting

Besides shielding the model from catastrophic overfitting, we observe that APART can also alleviate the robust overfitting. Specifically, as visualize the training curve of APART and PGD-10 in Figure 5, both APART and PGD-10 consistently get better training accuracy in the first 200 epochs. Meanwhile, after the first 30 epochs, both methods suffer from robustness drop against the PGD-20 attack. This phenomenon is referred as robust overfitting, and we can observe that APART suffers less from the robust overfitting. To better understand this phenomenon, we also calculated their perturbation strength gap to PGD-20, i.e., $\mathcal{G}_{\text{PGD-20},i}(\text{method}) = \mathcal{L}(\theta_{\text{method}}^{(i)}; \mathbf{x} + f_{\text{method}}(\theta_{\text{method}}^{(i)}, \mathbf{x}, y), y) - \mathcal{L}(\theta_{\text{method}}^{(i)}; \mathbf{x} + f_{\text{PGD-20}}(\theta_{\text{method}}^{(i)}, \mathbf{x}, y), y)$. We can find that both methods get larger perturbation strength gap in the later stage of training, and APART consistently gets a smaller gap. Intuitively, adaptive perturbation generator can prevents the model overfitting to a single type of perturbation, thus alleviating the robust overfitting.

## 6 RELATED WORK

### 6.1 ADVERSARIAL TRAINING

Goodfellow et al. (2014) first recognize the cause of the adversarial vulnerability to be the extreme nonlinearity of deep neural networks and introduced the fast gradient sign method (FGSM) to generate adversarial examples with a single gradient step. Madry et al. (2017) propose an iterative method based on FGSM with random starts, Projected Gradient Descent (PGD). PGD adversarial training is effective but time-consuming, and thus some recent work also pays attention to the efficiency of adversarial training. For example, Shafahi et al. (2019b) propose to update both the model parameters and image perturbations using one simultaneous backward pass. Zhang et al. (2019a) show that the first layer of the neural network is more important than other layers and make the adversary computation focus more on the first layer. Zheng et al. (2019) also improve the utilization of gradients to reuse perturbations across epochs. Wong et al. (2020) use uniform random initialization to improve the performance of FGSM adversarial training. APART improves the efficiency and effectiveness of adversarial training by factorizing the input perturbation as a series of perturbations. Previous methods only added the perturbation to input images, while APART adds perturbation to the input of residual blocks. Perturbations added to intermediate variables help improve the robustness, as discussed in Section 5.5.

### 6.2 ROBUSTNESS DROP

Wong et al. (2020) mention the robustness drop as overfitting and first identify a failure mode named as "catastrophic overfitting", which caused FGSM adversarial training to fail against PGD attacks. Rice et al. (2020) further explore the overfitting in other adversarial training methods, such as PGD adversarial training and TRADES. They observe that the best test set performance was achieved after a certain epochs and further training would lead to a consistent decrease in the robust test accuracy, and therefore explain it as "robust overfitting". Rice et al. (2020) show that robustness drop is a general phenomenon but they did not analyze its cause. Kim et al. (2021) asserts that catastrophic overfitting is caused by the fixed perturbation step size in single-step adversarial training, while we found PGD-2 may also suffer from catastrophic overfitting, even it does not fix the perturbation step size. In this work, we explore the nature of robustness drop in adversarial training and further propose APART to address the perturbation deterioration issue.

## 7 CONCLUSIONS AND FUTURE WORK

In this paper, we attempt to explore the mechanism behind catastrophic overfitting. As the common wisdom views the robustness drop as model overfitting, our analyses in Section 3 present a novel perspective and suggest that the other side of catastrophic overfitting is perturbation deterioration. Guided by our analyses, we propose APART, an adaptive adversarial training framework. APART parameterizes the perturbation initialization, factorizes the input perturbation into a series of perturbations (one for each layer in the neural networks), and progressively strengthens them during the training. In our experiments, APART not only successfully shields the model from catastrophic overfitting, but also achieves consistently performance improvements while maintaining roughly the same training efficiency with FGSM-style methods.

The major limitations of our method is that it can only be applied to residual networks, and it achieves faster training at the cost of some model robustness. There are several interesting directions to pursue in future work, including applying APART to general neural models, further improve the performance of APART by further strengthing the perturbation generator. Besides, we plan to explore the underlying mechanism of other phenomenons, like the trade-off between clean accuracy and robust accuracy.

**Reproducibility.** In this study, we conduct experiments on three public datasets, i.e., CIFAR-10, CIFAR-100, and ImageNet. We will release implementations for all methods and scripts for all experiments on GitHub, under the Apache-2.0 license.

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
