# OpenReview forum: "Perturbation Deterioration: The Other Side of Catastrophic Overfitting"
_ICLR.cc/2022/Conference — ICLR 2022 Submitted_

### Official Review · Reviewer_Bmpq · 2021-10-18

**Correctness:** 3
**Technical Novelty And Significance:** 2
**Empirical Novelty And Significance:** 2
**Recommendation:** 3
**Confidence:** 5

**Main Review:**

**Weaknesses and Questions**

1. The key argument in the paper for catastrophic overfitting is the deterioration of the augmented adversaries during the training time. However, while this has been supported with several empirical experiments, this was already known in literature. For example, the work by Hoki Kim et al (AAAI21) showed that the loss surface when catastrophic overfitting happens deviates from being linear. Since, FGSM is a closed form solution to the linearized loss, i.e. $\max_{\delta \in \mathcal{S}} \mathcal{L}(x,y) + \delta^\top \nabla_x \mathcal{L}(x,y)$ where $\mathcal{S}$ is the $\ell_\infty$ ball, this implies that the locally the loss function is far from linear. Therefore, the solution of FGSM very crudely approximates the loss surface and thereafter provides a worse approximation to the worst case adversary (original problem $\max_{\delta \in \mathcal{S}} \mathcal{L}(x+\delta,y)$). That is to say, FGSM solutions gets worse equivalently adversaries are deteriorating. To that end, I do not find the empirical experiments conducted in Figures 1 and 2 to provide any new insights that were not known before. It is merely a re-statement to what we already know about catastrophic overfitting. To that end, I do not find the first key component of this paper to be novel or insightful (the part on perturbation deterioration). In fact, this observation was the key element for the Hoki et al AAAI21 work of using stronger adversaries when the linearization as an approximation is worse.

2. The other key contribution to this paper is the algorithmic part. The paper does not report robust accuracy under larger $\epsilon$. As stated by the paper in section 5.2, F-FGSM, still suffers from catastrophic overfitting. This has also been observed by Gradalign "Understanding and Improving Fast Adversarial Training". To see this, one needs to evaluate F-FGSM trained models against a larger $\epsilon$, e.g. $\epsilon \ge 10/255$ (see Figure 1 left in their paper). This demonstrated that adding random noise in F-FGSM only helped against catastrophic overfitting for some choices of evaluated attack $\epsilon$. To that end, it is essential for this paper to conduct similar experiments as reported in Figure 1 of Andriushchenko et al to confirm whether catastrophic overfitting has been evaded for this paper.

3. I find it hard to understand the motivation of the per layer adversary perturbations. Can the authors comment, or maybe justify empirically on to why do one expect to improve against input adversary by augmenting per layer input adversaries?

4. The key motivation for the algorithmic contribution stems from the fact that one needs to adaptively control the strength of the adversaries during training. To elleviate this difficulty, the paper proposes to learn the step size along with the initialization for the FGSM step. I find the technical contribution here to be minimal. This replaces one problem with another. Instead of selecting the step size $\alpha$ for FGSM, this procedure is replaced to selecting a learning schedule for the optimzier learning $\alpha$. Thus, a similar nuisance to FGSM can occur where a choice of $\alpha$ could lead to catastrophic overfitting, a choice of scheduler for $\alpha$ could lead to a catastrophic overfitting. Thus, since there are no guarantees and that the key element here is replacing the tuning of a scalar to tuning a scheduler is not considered as a major contribution. I may be wrong or missing something here. I hope authors correct me on this.


5. Since the key contribution is the learning procedure (adaptive part) of adversary augmentation during training, I believe most of the experiments should have been conducted where only the adaptive element is presented at the input level. As a plus, one then considers that less motivated variant of per layer adaptive augmentation of adversaries. Currently, even with Table 6, it is very hard to understand whether there is enough improvement with the main claim of the paper. This is specifically that it seems that replacing a constant step size with a scheduler can actually still be worse that simple F-FGSM in complexity with almost similar robust accuracy.




**Algorithmic Typo?**
In Algorithm 1, are the updates in Lines 4 and 8 correct. This seems to be a constant function equal to $\epsilon$ and $1$, respectively, agnostically from the gradient step. I believe the paper meant to switch the order of max and min or the $-\epsilon$ and $\epsilon$ in min max operators and equivalently the $-1$ and $1$ for Line 8.

**Minor Comments**
1. Page 2, Line 5 just below Figure 1: "APART improve the initialization" >> "Improves".
2. Section 4, Page 4, Line 4: "Moreover, it employ" >> "employs".
3. Section 5.7, Line 2: "Specifically, as visualize the training" >> "we visualize ..".
4. Consider making the legend of Figure 5(c) bigger. It is currently hard to read.
5. Missing citation: "Understanding and Improving Fast Adversarial Training".



**Summary Of The Paper:**

**Summary**

The paper address the issue of catastrophic overfitting in robust training. Robust overfitting is often attributed to the fact that robust training tends at some point during the training to overfit to the underlying threat model augmenting the adversaries. This cases the test robust accuracy against the augmented training adversaries to be high while achieving 0% robust accuracy against stronger threat models. The paper provides new insights to potential other reasons to this phenomena. In particular, the paper argues that that at the same epoch when catastrophic overfitting happens, the quality of the augmented adversaries deteriorates fast and is no stronger than a random noise agumentation. The paper then goes about resolving this issue by strengthening the augmented adversaries during training by learning the hyperparameters of FGSM adversaries, namely the step size and the initialization. In addition, the paper augments the adversaries per Residual block, i.e. finding FGSM attacks per layer input. This approach is titled APART and shows competitive performance compared against existing methods in both final robust accuracy and training time efficiency. Experiments are conducted on CIFAR10, CIFAR100 and ImageNet datasets.

**Summary Of The Review:**

See the main review.

---

### Official Review · Reviewer_amVQ · 2021-11-02

**Correctness:** 3
**Technical Novelty And Significance:** 2
**Empirical Novelty And Significance:** 2
**Recommendation:** 5
**Confidence:** 5

**Details Of Ethics Concerns:**

No Ethics Concerns.

**Main Review:**

1. It is difficult to understand that catastrophic overfitting is related to the type of perturbation. I learned that Figure 1 tries to show the phenomenon that can support the authors' claim. However, it is difficult to determine whether the FGSM-based perturbation is similar to the random noise that leads to overfitting. Or, when overfitting occurs, the model worsens, making the FGSM-based perturbation look random. Besides,  the results in this paper are similar to an existing work [1]. The work [1] also considered changing the perturbation types to overcome the overfitting.

2. In Algorithm 1, why do the authors calculate the perturbation for each residual block? I think this is similar to the adversarial weight perturbation [2]. What's the difference? Furthermore, if we don't adopt the network with residual structure, what modification should be made in this place?

3. If we also use this adaptive attack method during evaluations, what will the experimental results be?

[1] Towards Understanding Fast Adversarial Training, arxiv.
[2] Adversarial Weight Perturbation Helps Robust Generalization.

**Summary Of The Paper:**

This paper focuses on an emerging topic in adversarial training, which is about catastrophic overfitting. First, the authors observe that such overfitting is related to the limited types of perturbations. So, the authors introduce an adaptive adversarial training method, called APART, which adjusts the parameters of the perturbation generator. The experiments can verify that the APART can achieve competitive results.

**Summary Of The Review:**

The observation results lack reasonable verification. And, the evaluation of adaptive attack is missing.

---

### Official Review · Reviewer_Z8eC · 2021-11-03

**Correctness:** 3
**Technical Novelty And Significance:** 2
**Empirical Novelty And Significance:** 3
**Recommendation:** 6
**Confidence:** 2

**Main Review:**

Strengths (motivation, correctness, clarity):

Writing: The paper is well motivated, its writing is clear and easy to understand.

Observation: The observation that the abrupt drop in robustness may be due to perturbation degrading to random noise (drastic distribution shift) is new, to my best knowledge. I think this is an interesting empirical observation.

Methodology: The proposed APART algorithm is sensible, with reasonable performance while being computationally advantageous. I appreciate that the authors performed an ablation study.

Experiments: The experiments validated the paper's claims. Although their methods don't outperform the stronger methods, they are much faster than them. The paper nicely adds to practitioners' toolbox between the two spectrums, from being fast to accurate.


Weakness (in depth analysis, claim rigor):

In depth analysis:
While the paper may be (among) the first that performs a thorough study of such co-occurrences, the intuition where the generated adversarial samples being too weak, can cause training failure, is not new in differentiable games. In GAN training, discriminator being too strong may lead to vanishing gradient for the generator and other phenomena [1] [2]. I think such related works (maybe ones that are more immediately relevant to adversarial training) can be discussed to better motivate the current work. The similarities and differences may be interesting to compare. The present work can benefit additionally by examining/hypothesizing how the random noise like perturbation suddenly occurred.

Claim rigor/theoretical discussion:
- As stated in my summary, adversarial training's learning (theoretic) framework differs from those in empirical risk minimization. Since a major contribution of the paper is that the catastrophic overfitting and perturbation deterioration are highly correlated, the readers can benefit from a more formal comparison of the identified learning, to IID learning, and adversarial learning when the perturbation distribution doesn't change too much (degrading to random noise). Note I'm not suggesting the authors to prove theorems, etc., just a more formal framing.

- In section 3.1, "not only look like random noise, but also behaves like random noise" - I'd suggest the authors perform some statistical testing to validate this claim.

---
After seeing others' reviews, while I do believe the main writing it clear, the authors may fail to justify the most crucial part of their contribution (how perturbation distribution and robustness are linked).


[1] Towards Principled Methods for Training Generative Adversarial Networks
[2] Unbalanced GANs: Pre-training the Generator of Generative Adversarial Network using Variational Autoencoder

**Summary Of The Paper:**

The paper made two contributions: 1. it identified the co-occurrence of abrupt robustness drop and perturbation degrading to (potentially) random noise; 2. it proposed a computationally faster adversarial training algorithm while attaining a good accuracy. Point 1. is interesting: it suggests previously identified "catastrophic overfitting" may be due to the perturbation in training time fails to mimic those in test time in distribution, while the adversarial training is being performed. This is different from classical empirical risk optimizations, where the training data do not depend on the training processes.

**Summary Of The Review:**

Overall, I believe this paper merits publication. The only drawbacks I can see are the lack of discussion and some minor claims being not fully rigorous. I'd raise my rating if the author can give a more in depth discussion and analysis on the relation between adversarial robustness drop and changes in perturbation distribution.

---
After seeing other reviews, I think clarifying the learning theoretic framework in the current work is necessary, otherwise, I'd lower my score. It is unclear whether the authors explicitly thought through this point. It can be my biased and overly optimistic interpretation.

---

### Official Review · Reviewer_996w · 2021-11-03

**Correctness:** 3
**Technical Novelty And Significance:** 2
**Empirical Novelty And Significance:** 2
**Recommendation:** 3
**Confidence:** 3

**Main Review:**

**Strengths**

1. The proposed method is simple yet effective with respect to the balance between generalization performance and computational costs.

2. Author analyzes the behavior of FGSM-style adversarial training and proposes a method from the lens of perturbation deterioration.

**Weaknesses**

1. As clearly described in the abstract, this paper mainly shows that “at the same epoch when the FGSM-trained model catastrophically overfits, it generated perturbations deteriorate into random noise”. However, except the causal direction, this claim is trivial since a poorly performed model returns random prediction and one does not need strong perturbation to change the prediction.
Besides, it seems the paper assume that the deteriorate perturbation *cause* catastrophic overfitting. For example, in abstract they claim “Intuitively, once the generated perturbations become weak and inadequate, models would be misguided to overfit those weak attacks
and fail to defend strong ones”. However, if the perturbation becomes simple random noise, it can not cause catastrophic overfitting since it means the adversarial training becomes standard supervised learning.

2. This paper compares APART with several existing computationally efficient methods (free-8, S+FSGM, and F+FSGM), however, connection to these works is not investigated. Especially, it is not clear whether the success of prior works can be seen as the prevention of perturbation deterioration or not as with the APART does.

3. Strengthen the perturbation using parametric adversarial examples is simple and nice. However, this paper lacks several prior works that use different parameterization to generate adversarial examples [1,2]. Connection to these works should be discussed.

[1] Baluja, Shumeet and Ian S. Fischer. “Adversarial Transformation Networks: Learning to Generate Adversarial Examples.” (2017)
[2] Xiao, Chaowei et al. “Generating Adversarial Examples with Adversarial Networks.” (2018)

4. There is no theoretical analysis, and therefore this is basically an empirical paper and requires strong empirical results to be accepted.  However, the performance improvement over prior computationally effective methods seems to be incremental in many settings (e.g. except PGD-20 in Cifar10). Besides, the discussion on the performance improvement is not enough.
Minor comments
I can not understand the meaning of Fig 1-a, as the color bar is grayscale yet the content is RGB.
Eq. 3 should maximize omega_alpha, not omega.


**Summary Of The Paper:**

This paper focuses on stable and effective adversarial training and improving generalization using it. This paper first investigates the phenomenon called catastrophic overfits in adversarial training, and provides a new hypothesis that the phenomenon occurs since the generated perturbations become too weak. Based on the hypothesis, this paper proposes APART, which parametrizes the adversarial example and trains it using gradient accent to prevent perturbation deterioration. Using several standard benchmark datasets, they show that APART prevents catastrophic overfitting and provides comparable performance with several existing methods (including PGD, Free-8, and variants of FGSM) in terms of generalization performance and computational costs.


**Summary Of The Review:**

Overall, I found the proposed method potentially interesting for fields in adversarial examples. However, the paper needs relatively large revision to fully support the claim and elaborate the novelty and significance of the work. The experimental results are ok but not strong enough to be accepted.

---

### Decision · Program_Chairs · 2022-01-20

**Decision:**

Reject

**Comment:**

The paper focuses on the Catastrophic Overfitting problem of adversarial training of FGSM. One reviewer gave a score of 6 and the other three reviewers gave negative scores. The authors failed to address or clarify (no rebuttal provided) how perturbation distribution and robustness are linked (four reviewers all agree on this). Other issues include unclear motivation, limited experiments validation, and lack of theoretical analysis. Thus, the current version of the paper cannot be accepted to ICLR.